# Knowledge, attitude and practice of home management of diarrhea among under-five children in East Africa: A systematic review and meta-analysis

**Biruk Beletew Abate**[1]*, **Alemu Birara Zemariam**[2], **Addis Wondimagegn**[3], **Gebremeskel Kibret Abebe**[3], **Freweyni Gebreegziabher Araya**[2], **Ayelign Mengesha Kassie**[4], **Molla Azmeraw Bizuayehu**[2]

**1** Assistant Professor in Pediatrics and Child Health Nursing, College of Health Science, Woldia University, Weldiya, Ethiopia, **2** MSc in Pediatrics and Child Health Nursing, College of Health Science, Woldia University, Weldiya, Ethiopia, **3** MSc in Emergency Medicine and Critical Care Nursing, College of Health Science, Woldia University, Weldiya, Ethiopia, **4** Assistant Professor in Adult Health Nursing, College of Health Science, Woldia University, Weldiya, Ethiopia

* birukkelemb@gmail.com

## Abstract

### Introduction

Diarrhea is particularly prevalent in low-income or marginalized populations because these groups have less access to clean water sources, hygienic conditions, and healthcare. Dehydration due to electrolyte and fluid loss is the main cause of deaths associated with diarrhea. An especially important factor in this death from dehydration is the caregivers' knowledge, attitude, and diarrhea management techniques. While a number of research have been done on managing diarrhea at home, the results tend not to be consistent. This systematic review and meta-analysis aimed to assess the pooled estimate of knowledge, attitude and practice of home-based management of diarrhea in East Africa.

### Methods

Preferred Reporting Items for Systematic Review and Meta-Analysis (PRISMA) guidelines was used to search articles from electronic databases (Cochrane library, Ovid platform (Medline, Embase, and Emcare), Google Scholar, CINAHL, PubMed, and institutional repositories in East Africa countries. The last search date was on 01/06/ 2023 Gregorian Calendar. The authors extracted year of publication, country, study design, knowledge level, attitude level and practice level of home-based management of diarrhea. A weighted inverse variance random-effects model was used to estimate the pooled prevalence of knowledge, attitude and practice of home-based management of diarrhea. Subgroup analysis was done by country, and sample size. Publication bias and sensitivity analysis were also done.

**Data Availability Statement:** The datasets used and/or analyzed during the current study is fully

available within the manuscript and supporting documents.

**Funding:** The author(s) received no specific funding for this work.

**Competing interests:** The authors have declared that no competing interests exist.

## Results

A total of 19 articles with (n = 7470 participants) were included for the final analysis. From the random-effects model analysis, the pooled prevalence of good practice, good knowledge and favorable attitude towards home based management of diarrhea in East Africa was found to be 52.62% (95% CI: 45.32%, 59.92%) (95% CI: I2 = 78.3%; p < 0.001), 37.44% (95% CI: 26.99%, 47.89%) (95% CI: $I^2$ = 89.2%; p < 0.001) and 63.05% (95% CI: 35.7%, 90.41%) (95% CI: $I^2$ = 97.8%; p < 0.001) respectively.

## Conclusion and recommendations

The level of good knowledge, attitude and practice of home based management of diarrhea in East Africa is found to be low. A collaborative effort from different stakeholders to enhance the knowledge, attitude and practice is needed to tackle the burden of diarrhea and its consequences.

## Introduction

Diarrhea is passing of three or more loose or liquid stools per day, or more frequently than is normal for the individual [1]. Diarrhea can be acute or chronic, and watery or bloody/dysentery. As diarrhea causes loss of body fluid and electrolytes it leads to dehydration which is the common cause of mortality, malnutrition, and other severe complications [2]. It is one of the leading causes of morbidity and mortality in children under the age of 5 all over the world [3,4]. Dehydration due to loss of fluids and electrolytes is estimated to be the main cause of diarrhea-related deaths causing 60%–70% of diarrhea-related deaths in children under the age of five [5–7]. Diarrheal diseases disproportionately affect people in locations with limited access to health care and safe water supply and proper sanitation, and belong to low-income or marginalized populations [8]. Factors of particular importance include caregivers' knowledge about the causes of diarrhea and the associated danger signs, on how to prevent dehydration during diarrheal episodes through the use of Oral Rehydration Salts (ORS) [9,10]. Therefore, it is strongly recommended to provide prompt rehydration therapy to children with diarrhea.

Globally in 2019, over 5 million children under the age of 5 years passed away and 484,000 of these deaths were due to diarrheal disease which could have been easily prevented and treated. Sub-Saharan African countries account more than half of the global burden of under-five mortality. Globally in 2016, almost 0.9 million deaths were related to unsafe drinking water, unsafe sanitation, and lack of hygiene; diarrhea accounted for over 470,000 deaths of under five years children in this year[11,12]. In the time span from 2000 to 2019, morbidity and mortality rate due to diarrhea has decreased from 12.6% to 9.1% by means of improvement in quality of water consumed, proper sanitation, vaccination, and adequate treatment coverage. Despite these efforts, diarrhea is still the leading cause of under-five mortality worldwide, especially in Africa [11,13,14]. In 2016, although worldwide safe water usage and sanitation reached 71% and 39% respectively, Africa is having a drawback on these needs (44% safely managed drinking-water services, and 21% safely managed sanitation services). Diarrhea in children can be managed at home before it becomes severe and life threatening. Home based management of diarrhea is a protocol developed to easily manage diarrhea and prevent the complications cost effectively. The management includes: increasing fluid and food intake including breast feeding, giving oral rehydration solution, and resting can suffice.

Inappropriate home management practice can worsen a child's condition and even causes death [15,16]. Effective therapies for diarrhea exist, but children in rural settings often do not have access to professional healthcare[17,18], and coverage of these interventions remains low in many developing countries [19]. People fail to adapt universally popular therapies like ORS in preventing dehydration [3]. Factors contributing to inadequate treatment of diarrhea include financial constraints, geographic inaccessibility to well-equipped healthcare facilities, and shortage of qualified health professionals and essentials pharmacologic treatment options, this in turn leads to diarrhea-related mortality [20]. Delivery of care through community health workers (CHWs) can increase coverage of specific treatments [21] and lead to substantial reductions in child mortality [22,23]. The World Health Organization (WHO) and the United Nations Children's Fund (UNICEF) recommend integrated community case management (iCCM) of diarrhea [24]. However, experience with large scale implementation of iCCM remains limited [25]. Few rigorous assessments of iCCM implementation have been conducted, and the limited evidence on quality of care is mixed [26–29].

Family plays the major role in the treatment and survival of children with diarrhea. For effective management of diarrhea at home, caregivers should have the basic knowledge and good attitude. Their role is vital in health promotion, disease prevention, and patient care. Proper home-based management can reduce morbidity and mortality due to diarrhea in children under the age of five. There are limited studies which investigate the level of home-based diarrhea management practice among the caregivers of children under the age of 5. In Africa different studies have been conducted regarding the knowledge, attitude, and practice of home-based management of diarrhea, but they lack consistency and results are variable; knowledge level ranges from 36.6% to 67%, attitude 45.1% to 94.4%, and practice 12% to 58%. As per the investigators' knowledge, there are no systematic reviews or meta-analyses was done to address this inconsistent data from East Africa. This systematic review and meta-analysis aimed to assess the pooled estimate of knowledge, attitude and practice of home based management of diarrhea in East Africa.

## Methods

### Search strategy

This systematic review and meta-analysis review assessed studies that provide data on the knowledge, attitude and practice of home-based management of diarrhea and /or its determinants in context of East Africa context. We searched these articles from the following databases: Cochrane library, Ovid platform (Medline, Embase, and Emcare), Google Scholar, CINAHL, PubMed, and institutional repositories in East Africa countries on 01/06/2023 Gregorian Calendar. The search in all database included keywords that are the combinations of population, condition/outcome, and context. A snowball searching for the references of relevant papers for linked articles was also performed. The following search map was applied: "knowledge", "awareness", "attitude", "perception" "practice", "home-based", "community-based", "management", "Diarrhea", "East-Africa" on PubMed database. These search terms were further paired with the names of East-African countries. Using those key terms, we used the Boolean operator "OR" (to connect key terms/phrases within the same concept), "AND" (to connect key terms /phrases between two concepts), and "NOT" to filter out. In addition, we used truncation (*), adjacency searching (**ADJn**), and wildcard symbols to find variations in spelling and variant word endings on the Ovid databases. Moreover, we applied relevant limits (filters) such as a limit to human studies only [30–34].The sample search strategy for Medline is attached in the appendix section of this manuscript.

## Study selection and screening

The retrieved studies were exported to Endnote version 8 reference managers to remove duplicate studies. Two investigators (BB and MA) independently screened the selected studies using article's title and abstracts before retrieval of full-text papers. We used pre-specified inclusion criteria to further screen the full-text articles. Disagreements were discussed during a consensus meeting with other reviewers (AB and AW) for the final selection of studies to be included in the systematic review and meta-analysis [35]. The retrieved studies were imported in covidence platform to remove duplicate studies and to do the whole screening process.

## Inclusion and exclusion criteria

**Inclusion criteria.** **Population:** studies conducted on under-five children
**Study design:** All observational studies
**Outcome:** Studies that assessed the level of knowledge, attitude and/or practice of home based management of diarrhea
**Study Area:** Studies conducted in East-Africa context.
**Publication year**: All articles published before 01/06/ 2023 were considered
**Exclusion criteria:** Citations without abstract and/or full text, anonymous reports, editorials, and qualitative studies were excluded from the analysis. Studies conducted on children more than five years old and on adult were excluded.

## Quality assessment

The authors appraised the quality of the studies by using the Joanna Briggs Institute (JBI) quality appraisal checklist [36]. There was a team of four reviewers and the papers were split amongst the team. Each paper was then assessed by two reviewers and any disagreements were discussed with the third and the fourth reviewers. Studies were considered as low risk or good quality when it scored 4 and above for all designs, whereas the studies which scored 3 and below were considered as high risk or poor quality and excluded.

## Data extraction

The authors developed a data extraction form on the excel sheet and the following data were extracted for eligible studies: year of publication, country, study design, knowledge level, attitude level and practice level of home based management of diarrhea. The data extraction sheet was piloted using 4 papers randomly, and it was adjusted after piloted the template. Two authors (BB and GK) extracted the data using the extraction checklist. The third and fourth (AW and FG) authors checked the correctness of the data independently. Any disagreements between reviewers were resolved through discussions with third and fourth reviewers when required [37,38]. The mistyping of data was resolved through crosschecking with the included papers.

## Synthesis of results

The authors transformed the data to STATA 17 for analysis after it was extracted in an excel sheet considering knowledge level, attitude level and practice level of home based management of diarrhea. The authors pooled the overall prevalence of knowledge, attitude and practice using a random effect model. We examined the heterogeneity of effect size using the Q statistic and the $I^2$ statistics. In this study, the $I^2$ statistic value of zero indicates true homogeneity, whereas the value 25%, 50%, and 75% represented low, moderate and high heterogeneity, respectively. Subgroup analysis was done by the study country. Sensitivity analysis was

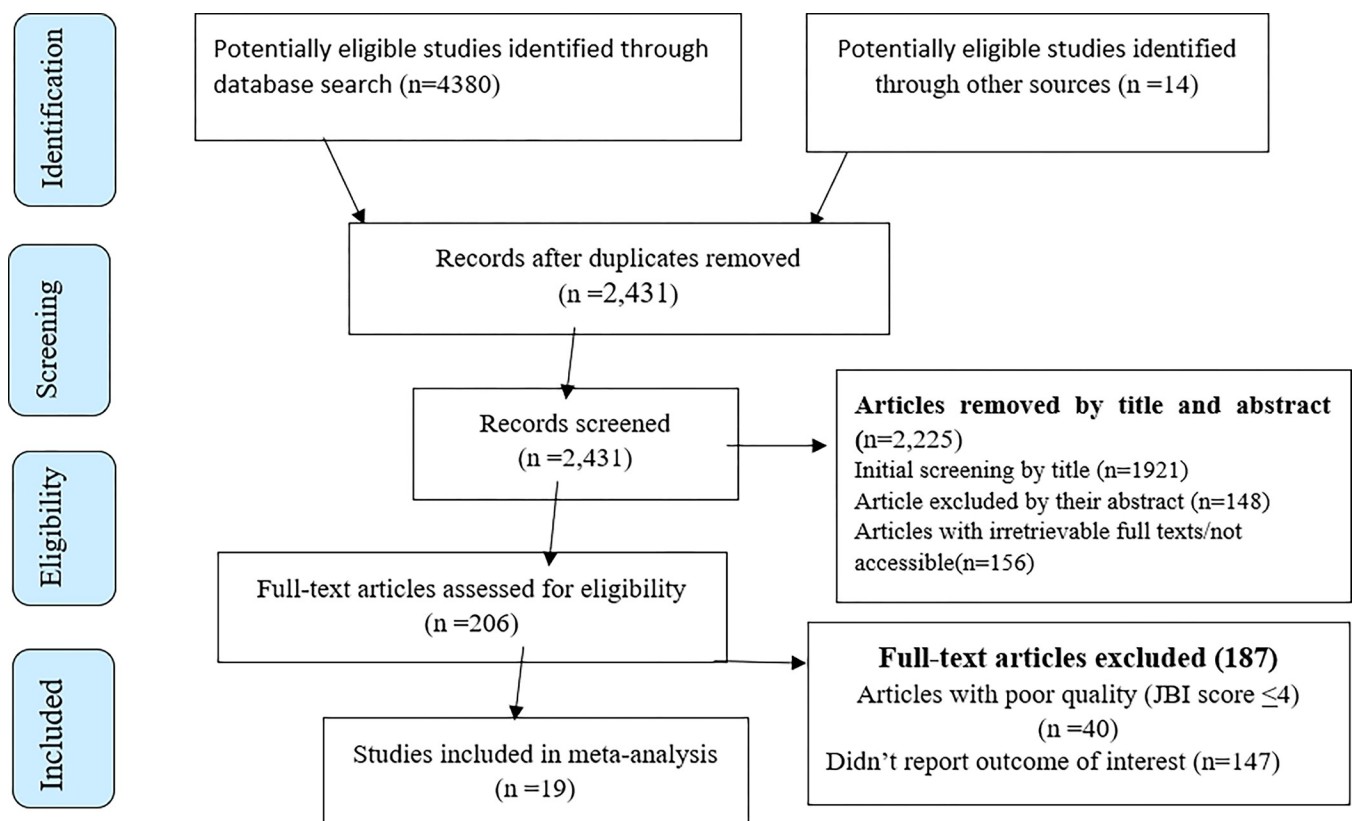

**Fig 1. PRISMA–adapted flow diagram showed the results of the search and reasons for exclusion.**

employed to examine the effect of a single study on the overall estimation. Publication bias was checked by the funnel plot and more objectively through Egger's regression test.

## Results

A total of 4394 studies were identified; 4380 from different databases and 14 from other sources. After duplication removed, a total of 2,431 articles remained (1963 removed by duplication). Finally, 206 studies were screened for full-text review, and 19 articles with (n = 498406 patients) were included for the final analysis (Fig 1).

### Characteristics of included studies

Table 1 summarizes the characteristics of the 19 included studies in this systematic review and meta-analysis. Twelve studies were from Ethiopia, [32–41] three from Kenya, one from Rwanda, one from South Sudan, and two from Uganda (Table 1).

### Knowledge level on home-based management of diarrhea

Regarding knowledge level of participants on home-based management of diarrhea, the pooled prevalence of good knowledge from the random-effects model analysis in Africa was found to be 52.62% (95% CI: 45.32%, 59.92%) (95% CI: $I^2$ = 78.3%; p < 0.001) (Fig 2). **S**ubgroup analysis was done through stratified by country. Based on this, the pooled prevalence of good knowledge was revealed 55.78% (47.77%, 63.79%) in Ethiopia, 42.74% (33.65%, 51.83%) in Kenya, and 44.4% (14.16%, 74.64%) in Rwanda (Fig 3).

**Table 1. Distribution of included studies on the pooled prevalence knowledge level, attitude level and practice level of home based management of diarrhea, 2023.**

| Sr no | Author | Year | Country | Design | Sample size | Knowledge | Practice | Attitude |
|---|---|---|---|---|---|---|---|---|
| 1 | Amare D, et al | 2014 | Ethiopia | Cross sectional | 846 | 63.6 | 45.9 | |
| 2 | Bogale Kassahun Desta et al. | 2017 | Ethiopia | Cross sectional | 378 | 56.2 | 37.6 | |
| 3 | Terefe Dodicho et al | 2016 | Ethiopia | Cross sectional | 654 | 67 | 47.2 | |
| 4 | Workie et al | 2018 | Ethiopia | Cross sectional | 295 | 65.2 | 58 | |
| 5 | Bitew et al. | 2015 | Ethiopia | Cross sectional | 845 | 49.3 | | |
| 6 | Merga N and Alemayehu T et al | 2010 | Ethiopia | Cross sectional | 232 | 37.5 | | |
| 7 | Bethelhem Shewangizaw et al | 2023 | Ethiopia | Cross sectional | 238 | 36.6 | | |
| 8 | Owiti Beatrice Anyango et al | 2018 | Kenya | Cross sectional | 394 | 39.2 | | |
| 9 | Leah Wambui Gathogo et al | 2021 | Kenya | Cross sectional | 345 | 47.8 | | |
| 10 | Ndayisaba Archange et al | 2019 | Rwanda | Cross sectional | 160 | 44.4 | 47.5 | |
| 11 | Nalubwama S et al | 2021 | Uganda | Cross sectional | 246 | | 36 | |
| 12 | Stephen B et al. | 2016 | Uganda | Cross sectional | 367 | | 21.8 | |
| 13 | Peterson M etal | 2017 | Kenya | Cross sectional | 366 | | 17 | |
| 14 | Aneeqa M etal | 2022 | South Sudan | Cross sectional | 158 | | 12 | |
| 15 | Gemechu T etal | 2022 | Ethiopia | Cross sectional | 335 | | 59 | |
| 16 | Bethelhem Shewangizaw et al. | 2023 | Ethiopia | Cross sectional | 238 | | | 55.5 |
| 17 | Abera and Assefa et al | 2018 | Ethiopia | Cross sectional | 233 | | | 94.4 |
| 18 | Workie, Sharifabdilahi et al. | 2018 | Ethiopia | Cross sectional | 295 | | | 45.1 |
| 19 | Bitew, Gete et al. | 2017 | Ethiopia | Cross sectional | 845 | | | 54.8 |

**Publication bias.** A funnel plot showed symmetrical distribution. The Egger's regression test-value was 0·149, which indicated that, the absence of publication bias. As a result, we didn't conduct trim and fill analysis (S1 Fig). **Sensitivity analysis:** We also employed a leave-one-out sensitivity analysis to identify the potential source of heterogeneity in the analysis of the pooled magnitude of knowledge on home based management of diarrhea. The results of this sensitivity analysis showed that the findings were not dependent on a single study (S2 Fig).

## Practice level on home based management of diarrhea

The pooled prevalence of good practice from the random-effects model analysis in East Africa was found to be 37.44% (95% CI: 26.99%, 47.89%) (95% CI: $I^2$ = 89.2%; p < 0.001) (Fig 4). **Sub**-group analysis was done through stratified by country. Based on this, the pooled prevalence of good practice was found to be 47.83% (42.04%,53.61%) in Ethiopia, 17.00% (9.44%,24.56%) in Kenya, 47.45% (16.95%, 78.05%) in Rwanda, 25.00% (13.37%, 36.64%) in Uganda, and 12.00% (1.10%, 25.10%) in South Sudan (Fig 5).

**Publication bias.** A funnel plot showed symmetrical distribution. The Egger's regression test-value was 0·965, which indicated that, the absence of publication bias. As a result, we did not conduct trim and fill analysis (S3 Fig). **Sensitivity analysis:** We also employed a leave-one-out sensitivity analysis to identify the potential source of heterogeneity in the analysis of the practice of home based management of diarrhea. The results of this sensitivity analysis showed that the findings were not dependent on a single study (S4 Fig).

## Level of attitude towards home based management of diarrhea

The pooled prevalence of favorable attitude from the random-effects model analysis in Africa was found to be 63.05% (95% CI: 35.7%, 90.41%) (95% CI: $I^2$ = 97.8%; p < 0.001) (Fig 6).

**Publication bias.** A funnel plot showed symmetrical distribution. The Egger's regression test-value was 0·965, which indicated that, the absence of publication bias. As a result, we

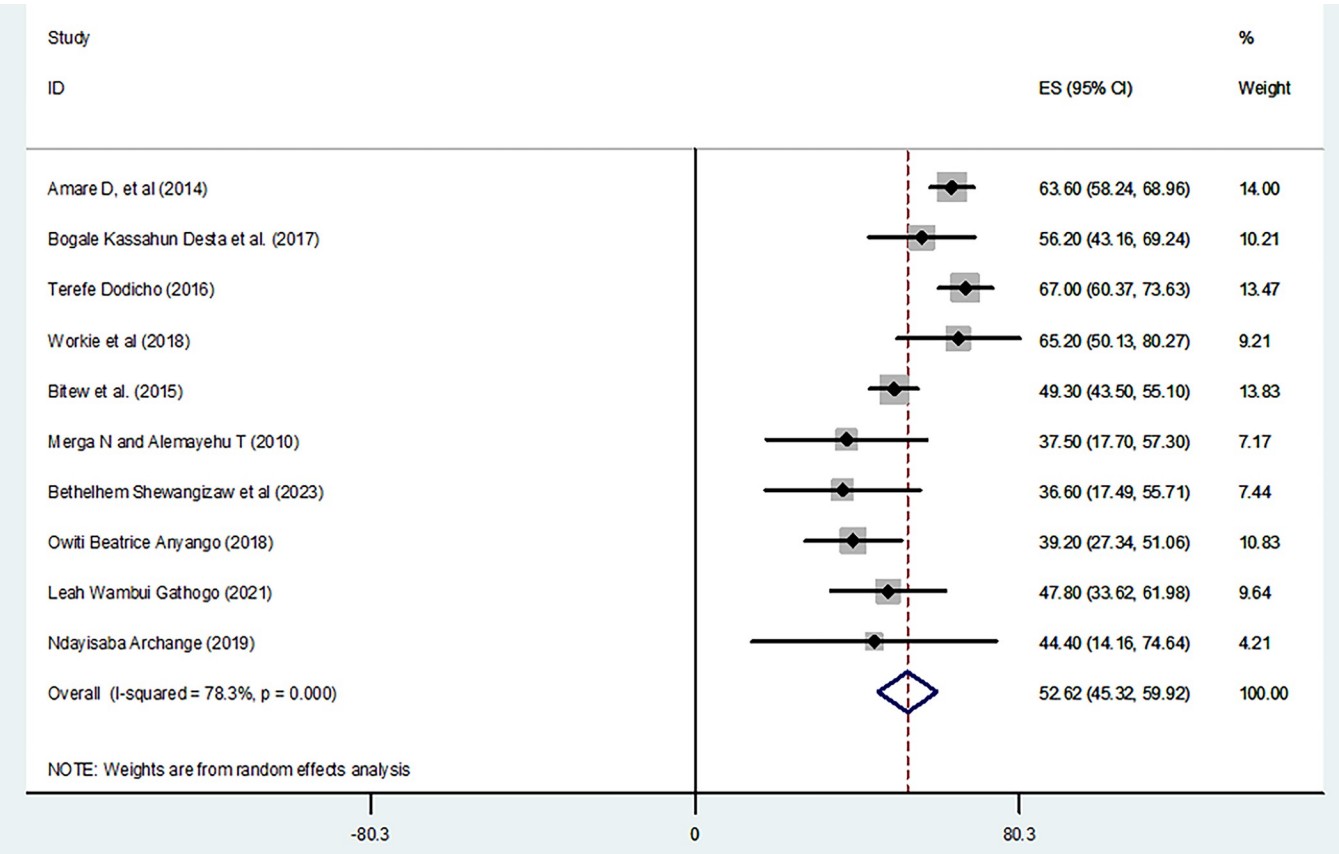

**Fig 2. Forest plot shows knowledge level on home based management of diarrhea in East Africa.**

didn't conduct trim and fill analysis (S5 Fig). **Sensitivity analysis:** We also employed a leave-one-out sensitivity analysis to identify the potential source of heterogeneity in the analysis of the attitude towards home based management of diarrhea. The results of this sensitivity analysis showed that the findings were not dependent on a single study (S6 Fig).

## Discussion

This systematic review and meta-analysis included 19 studies to assess the pooled estimate of knowledge, attitude and practice of home-based management of diarrhea in East Africa which were published from 2010–2021. As far as authors' knowledge is concerned, the study is the first of its kind in the aforementioned WHO region among under-five children. Accordingly, the finding revealed the pooled level of good knowledge, attitude and practice of caregivers on home-based management of diarrhea 52.62%, 60% and 37.4%, respectively.

The level of good knowledge varies from 42.74% in Kenya to 55.70% in Ethiopia. Similarly, the practice of caregivers on homebased management of diarrhea ranged from 12% South Sudan to 47.83% in Ethiopia. Studies have been limited about attitude towards homebased management of diarrhea in other nations of East Africa except Ethiopia, where the pooled good level of attitude is 63.05%. The finding was lower than the previous study in Afghanistan knowledge (84.8%), attitude (73.6%) and practice (80.7%) [41]. This discrepancy might be due to the fact that the background characteristics such as educational status, residence, and other socio economic parameters of the study participants in Afghanistan were better than that of

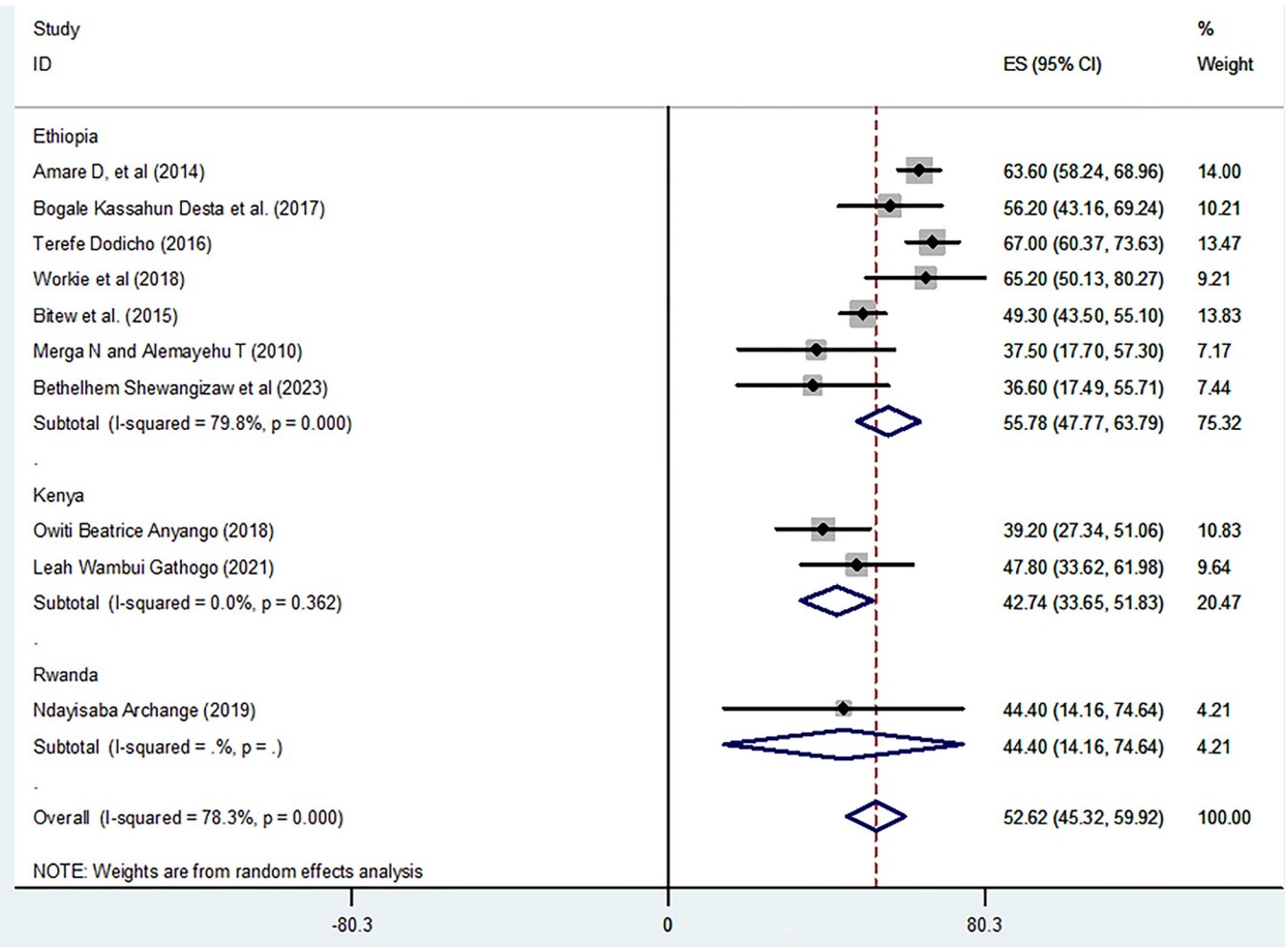

**Fig 3. Subgroup analysis knowledge level on home based management of diarrhea in by country East Africa.**

study participants in East Africa [41]. Previous studies also indicated that educational level, occupation and monthly income of the mothers' show association with the knowledge and practice whereas occupation and income level influence attitude of mothers towards home based management of diarrhea [42].

The level of good knowledge of caregivers in the current study was in line with other previous studies conducted in Nigeria (59.2%) [43], Yemen (51.2%) [44], and India (58.7%) [45]. This could be the result of various initiatives carried out by various governmental and nongovernmental organizations, such as the World Health Organization (WHO), which offered a comprehensive community health seeking behavior through a health task force comprising health extension workers and various healthcare providers. This led to the creation of a common understanding regarding the home management of diarrhea among various populations and nations. However, the current finding was higher than a study conducted in Namibia (36%) [46]. This difference could be attributed to the disparity in socio-economic and sociodemographic characteristics of the included population, such as the educational status of the included caregivers. Besides, this difference might be due to the sample size and outcome measurement variation across the study. On the other hand, the current finding was lower than studies conducted in Nigeria (95.4%) [47] and Pakistan (72.1%) [48]. Since the study in

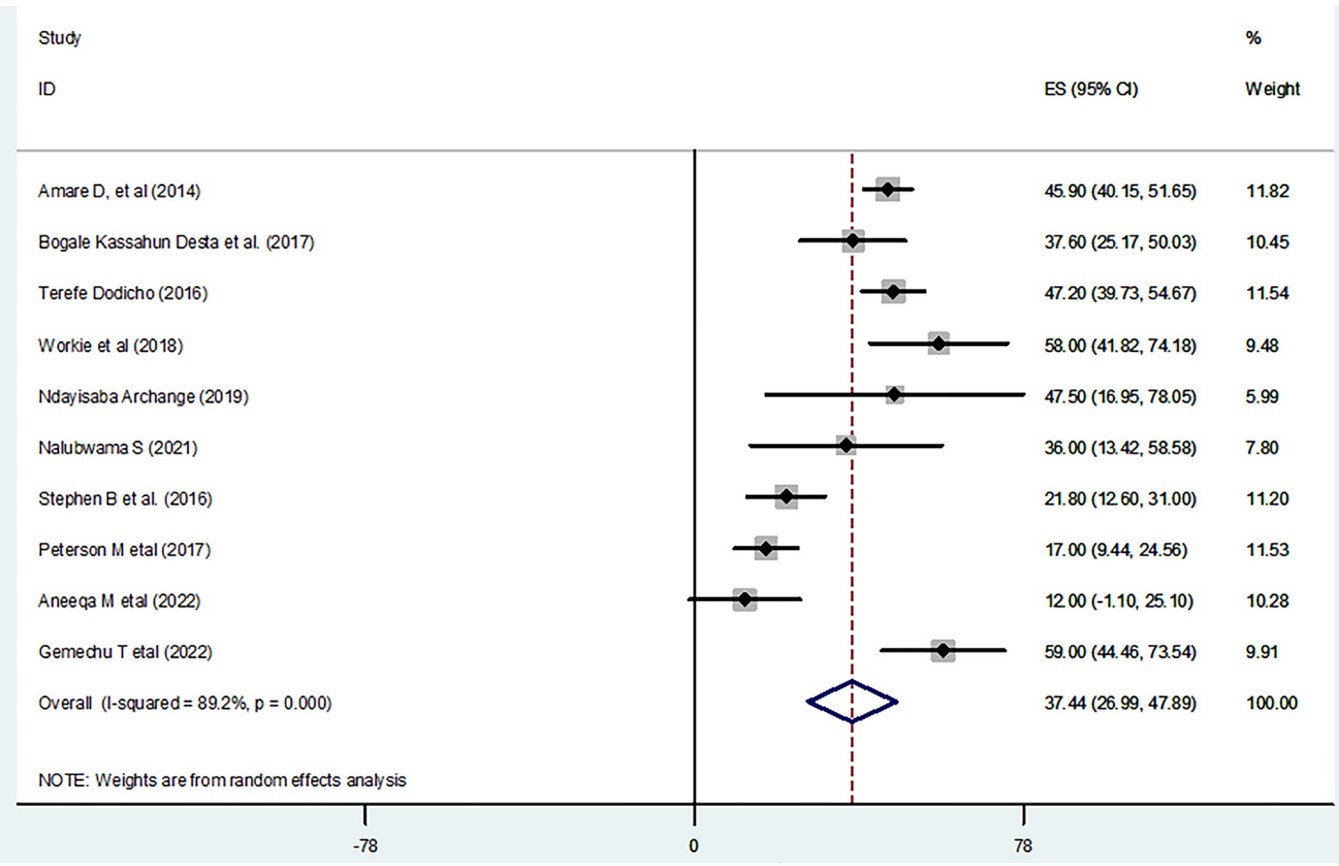

**Fig 4. Forest plot shows practice level of home based management of diarrhea in East Africa.**

Pakistan was done among urban slum communities, which are likely to have higher exposure to various health information and a good level of understanding of home-based management of diarrhea, one possible explanation for this difference could be population variance.

The poor level of knowledge, attitude and practice of home based management of diarrhea in East Africa might somewhat be influenced by their living situation, level of economic development, and dissemination of health information [49]. Furthermore, most African population were settled in rural areas where they cannot be accessed for delivery information and educations services on health care seeking behaviors of the community. The disparity, however, might be explained by the low rates of literacy and public awareness raised by formal means such as the media and other kinds of communication or advertising, which tend to reinforce prevailing cultural norms and beliefs in the community. The result suggests that in order to improve home-based management of diarrhea, fair and equal distribution of health-related information is necessary.

Furthermore, in subgroup analysis stratified by countries of included studies indicated that the pooled prevalence of good knowledge was higher in Ethiopia (55.78%) than other East African countries such as Kenya (42.74%) and Rwanda (44.4%). This discrepancy might be because Ethiopia has implemented community-based health extension program. Under this program, health extension workers are making home to home visits on regular bases to support families in accessing basic health services and to give home-based health education as well as other promotion services. The role of non-governmental organization (local and

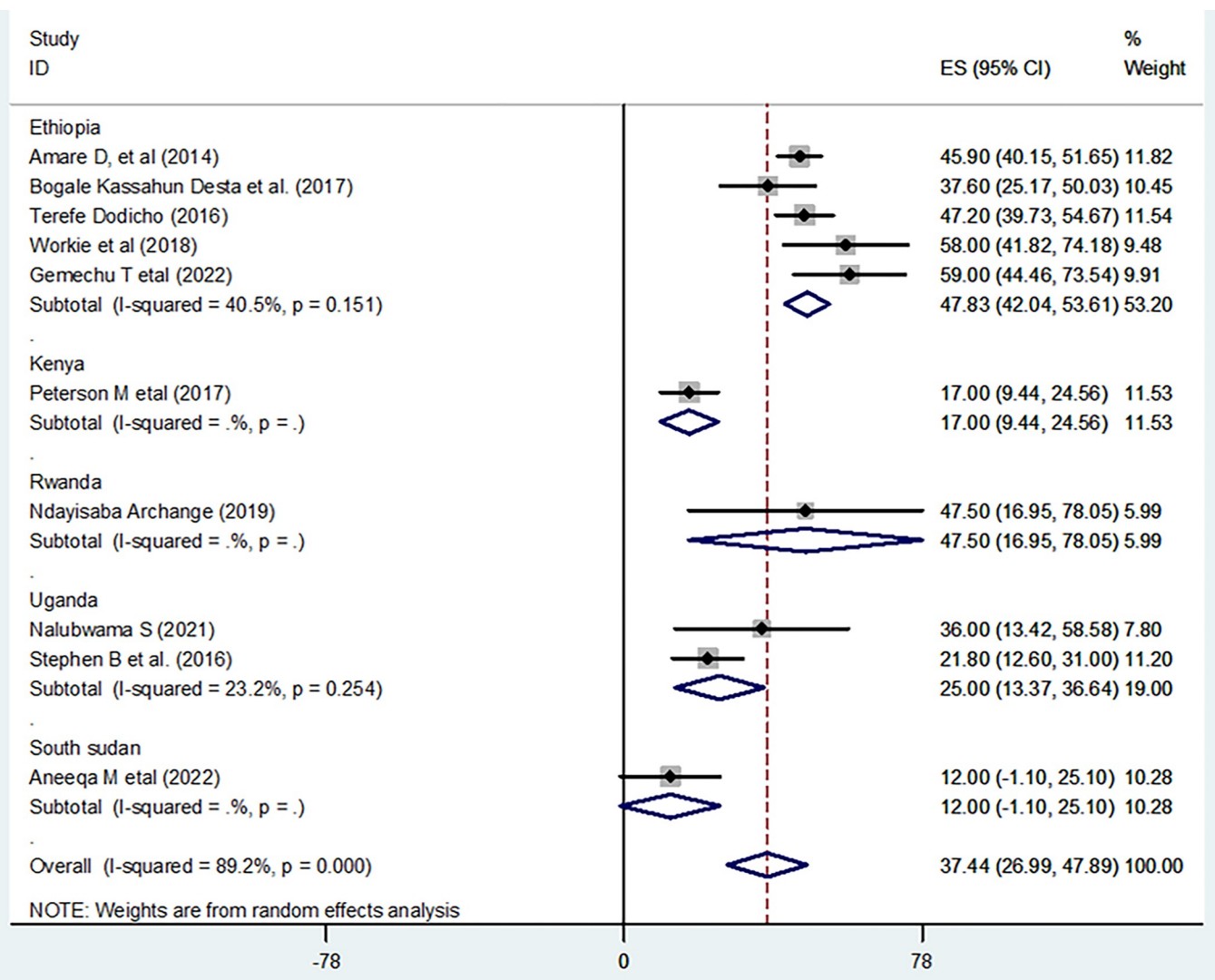

**Fig 5. Subgroup analysis shows practice level of home based management of diarrhea by country in East Africa.**

international) effort on strengthening health extension program implementation and support for primary health care system has been also appreciable in Ethiopia. Although this study provided a pooled data on the knowledge, attitude and practice of home-based management of diarrhea in East Africa, the authors couldn't able to find studies from some countries in East Africa which could affect the generalizability of the finding.

## Conclusion and recommendations

The level of good knowledge, attitude and practice of home-based management of diarrhea in East Africa is found to be low compared to previous study findings and WHO recommendation. A collaborative effort from different stakeholders to enhance the knowledge, attitude and practice is needed to tackle the burden of diarrhea and its consequences. This will in turn helps to achieve the sustainable developmental goal regarding reducing children mortality.

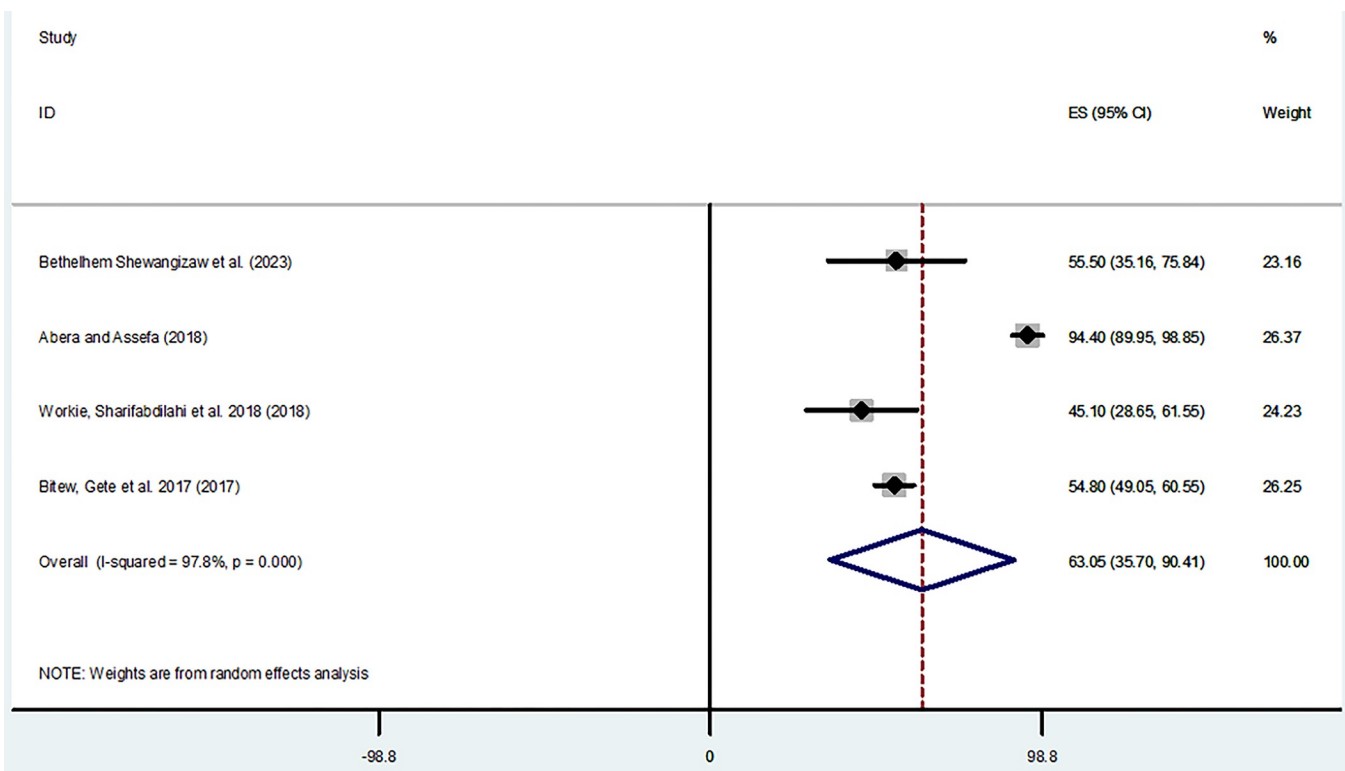

**Fig 6. Forest plot shows attitude level towards home based management of diarrhea in East Africa.**

## Supporting information

**S1 Checklist. PRISMA 2020 checklist.**
(PDF)

**S1 Fig. Shows publication bias knowledge on home based management of diarrhea in by country East Africa.**
(DOCX)

**S2 Fig. Shows sensitivity analysis of knowledge on home based management of diarrhea in by country East Africa.**
(DOCX)

**S3 Fig. Shows publication bias practice of home based management of diarrhea in by country East Africa.**
(DOCX)

**S4 Fig. Shows sensitivity analysis of practice of home based management of diarrhea in by country East Africa.**
(DOCX)

**S5 Fig. Shows publication bias attitude towards home based management of diarrhea in by country East Africa.**
(DOCX)

**S6 Fig. Shows sensitivity analysis on attitude towards home based management of diarrhea in by country East Africa.**
(DOCX)

## Author Contributions

**Conceptualization:** Biruk Beletew Abate, Addis Wondimagegn, Molla Azmeraw Bizuayehu.

**Data curation:** Biruk Beletew Abate, Alemu Birara Zemariam, Freweyni Gebreegziabher Araya, Ayelign Mengesha Kassie, Molla Azmeraw Bizuayehu.

**Formal analysis:** Biruk Beletew Abate, Alemu Birara Zemariam.

**Funding acquisition:** Biruk Beletew Abate.

**Investigation:** Biruk Beletew Abate.

**Methodology:** Biruk Beletew Abate, Freweyni Gebreegziabher Araya, Ayelign Mengesha Kassie.

**Project administration:** Biruk Beletew Abate.

**Resources:** Biruk Beletew Abate, Freweyni Gebreegziabher Araya, Ayelign Mengesha Kassie, Molla Azmeraw Bizuayehu.

**Software:** Biruk Beletew Abate, Addis Wondimagegn.

**Supervision:** Biruk Beletew Abate, Gebremeskel Kibret Abebe, Ayelign Mengesha Kassie.

**Validation:** Biruk Beletew Abate, Alemu Birara Zemariam, Addis Wondimagegn, Gebremeskel Kibret Abebe, Ayelign Mengesha Kassie, Molla Azmeraw Bizuayehu.

**Visualization:** Biruk Beletew Abate, Alemu Birara Zemariam, Addis Wondimagegn, Gebremeskel Kibret Abebe, Freweyni Gebreegziabher Araya, Molla Azmeraw Bizuayehu.

**Writing – original draft:** Biruk Beletew Abate, Alemu Birara Zemariam, Addis Wondimagegn, Gebremeskel Kibret Abebe, Freweyni Gebreegziabher Araya, Ayelign Mengesha Kassie, Molla Azmeraw Bizuayehu.

**Writing – review & editing:** Biruk Beletew Abate, Alemu Birara Zemariam, Addis Wondimagegn, Gebremeskel Kibret Abebe, Freweyni Gebreegziabher Araya, Ayelign Mengesha Kassie, Molla Azmeraw Bizuayehu.

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
