## [Decision Letter · Decision Letter 0]

13 Oct 2023

PONE-D-23-27980Knowledge, attitude and practice of home management of diarrhea among children in East AfricaPLOS ONE

Dear Dr. Abate,

Thank you for submitting your manuscript to PLOS ONE. After careful consideration, we feel that it has merit but does not fully meet PLOS ONE’s publication criteria as it currently stands. Therefore, we invite you to submit a revised version of the manuscript that addresses the points raised during the review process.

ACADEMIC EDITOR: Please make sure to address all comments raised by the reviewers and proof-read the article for grammatical and English errors. Please make sure to look into the reviewers have comments below as well as comments on the manuscript in attachments provided below.

We look forward to receiving your revised manuscript.

Kind regards,

Sidhant Ochani, MBBS

Academic Editor

PLOS ONE

Journal Requirements:

-DOI: 10.1177/20503121221095727

-https://doi.org/10.1016/j.vhri.2021.03.005

In your revision ensure you cite all your sources (including your own works), and quote or rephrase any duplicated text outside the methods section. Further consideration is dependent on these concerns being addressed.

5. Please amend the manuscript submission data (via Edit Submission) to include authors Gebremeskel Kibret Abebe and Freweyni Gebreegziabher Araya.

6.Please amend your authorship list in your manuscript file to include authors Biruk Kibret Beletew and Biruk Gebreegziabher Beletew. 

Additional Editor Comments:

The manuscript requires revision, since it has many errors and should be review-edited in proficient English for easy readability. The title of the manuscript should include the study type: Knowledge, attitude and practice of home management of diarrhea among children in East Africa; a Systematic Review and Meta-analysis and register your study protocol at PROSPERO. Additionally, please refer to the submission and publication criteria in the journal's guidelines for Systematic Reviews and Meta-Analysis.

Reviewers' comments:

Reviewer's Responses to Questions

**Comments to the Author**

1. Is the manuscript technically sound, and do the data support the conclusions?

Reviewer #1: Partly

Reviewer #2: Yes

Reviewer #3: Partly

2. Has the statistical analysis been performed appropriately and rigorously? 

Reviewer #1: I Don't Know

Reviewer #2: Yes

Reviewer #3: Yes

3. Have the authors made all data underlying the findings in their manuscript fully available?

Reviewer #1: Yes

Reviewer #2: Yes

Reviewer #3: Yes

4. Is the manuscript presented in an intelligible fashion and written in standard English?

Reviewer #1: No

Reviewer #2: No

Reviewer #3: Yes

5. Review Comments to the Author

Reviewer #1: 1. The study characteristics table is on a different topic, and seemed to be inserted by mistake.

2. Subgroup analysis missing for Uganda and South Sudan, funnel plot missing from manuscript and supplementary materials.

3. All data included in the manuscript is fully available.

4. Not at all, authors need to revise their grammatical errors tremendously.

Reviewer #2: The authors conducted a systematic review of publications on the pooled knowledge, attitude and practices of caregivers of children on home-based diarrhea management in East African countries. The study determined that the values were 52.62%, 60% and 37.4%, respectively. Thus, the knowledge, attitude and practices of home-based management of diarrhea in East Africa were found to be low in contrast to some previous reports and despite WHO recommendation. Although, this is a somewhat important summary, it does not seem that the analysis contributes much (if any) novel knowledge or resolves any existing controversies/inconsistencies in the field. Also, a thorough revision throughout the manuscript is needed to ensure more concise, proper and informative language/style. Currently, it is a struggle to read.

Specific comment:

Please clarify why “Finally, 206 studies were screened for full-text review, and 15 articles with (n = 498406 patients) were included for the final analysis (Fig. 1).” (P8). What was the basis for non-selecting other 191 articles? Why were they excluded and only 15 were selected? What is the meaning of ‘quality reasons’ and ‘Didn’t report outcome of interest’ shown in the diagram.

Some examples of errors, poor style, typos. Keep in mind that almost every sentence needs some level of revision.

P1: “Diarrheal disproportionately affects locations with poor access..” → “Diarrheal disease disproportionately affects locations with limited access..” Can’t say poor access. Same sentence needs to be corrected on P2.

P1: “Factors of particular importance include care givers knowledge…”

Please revise throughout to avoid saying: “estimate the pooled estimates of knowledge”. Poor style.

Please replace “care givers knowledge” with ‘caregiver knowledge’ throughout.

P2: “From the random-effects model analysis The pooled prevalence of good practice..” Do not randomly capitalize words inside a sentence.

P2: “The level of good knowledge, attitude and practice of home based management of diarrhea in East Africa is found to low” → “The level of good knowledge, attitude and practice of home based management of diarrhea in East Africa is found to be low”

P3: “Sub-Saharan Africa countries take more than half of the global burden of under-five mortality.” → “Sub-Saharan African countries account more than half of the global burden of under-five mortality.”

P3: “Diarrhea in children can be managed at home before it becomes sever and problematic.” → “Diarrhea in children can be managed at home before it becomes severe and problematic.” I would also replace ‘problematic’ with ‘life-threatening’

P3: “…often do not have access to formal healthcare”… What is ‘formal healthcare’? ‘Professional healthcare’

P3: “The level of home management practice of diarrhea is poor.” Please provide a reference to support this statement.

P3: “. Similarly, their practice to use universal popular ORS in preventing dehydration due to diarrhea is also very low[3]” Whose is ‘their’? Who are you referring to with this pronoun?

P4: “Family plays the major role in the treatment and surviving chance of a child with diarrhea.” → “Family plays the major role in the treatment and survival of children with diarrhea.”

P4: “In Africa different studies have conducted regarding Knowledge, attitude, and practice on home based management of diarrhea and they lack consistency (knowledge ranges from 36.6% to 67%, attitude 45.1% to 94.4%, and practice 12% to 58%).” → “In Africa different studies have been conducted regarding the knowledge, attitude, and practice of home-based management of diarrhea but they lack consistency (knowledge ranges from 36.6% to 67%, attitude 45.1% to 94.4%, and practice 12% to 58%).”

Unnecessary/imp[roper capitalization, improper use of conjunctions, missing articles and verbs are a big issues throughout the manuscript.

P4: “As per the investigators knowledge there are no systematic review and meta-analysis done to address the inconsistence reports from Africa.” → “As per the investigators knowledge, no systematic review or meta-analysis was done to address this inconsistency (or these inconsistent data) from Africa.”

P5: “We searched these articles from the following databases: Cochrane library, Ovid platform (Medline, Embase, and Emcare), Google Scholar, CINAHL, PubMed, and institutional repositories in East Africa countries on 01/06/ 2023 G. C” → “We searched these articles from the following databases: Cochrane library, Ovid platform (Medline, Embase, and Emcare), Google Scholar, CINAHL, PubMed, and institutional repositories in East African countries on 01/06/ 2023 G. C”

P8: Either omit ‘After duplication removed’ or “(1963 removed by duplication)”, having both in the sam sentence is redundant.

P17: “A concerted effort is needed to put an end to all avoidable infant and child deaths under the age of five in order to achieve the sustainable development goals (SDG) aim of decreasing under-five mortality to 25 per 1000 live births[50].” → “A concerted effort is needed to put an end to all avoidable deaths of infants and children under the age of five in order to achieve the sustainable development goals (SDG) aim of decreasing under-five mortality to 25 per 1000 live births[50].” I would omit ‘under the age of five’ The aim should be to prevent all avoidable death for children of all ages.

Reviewer #3: Well done on an important topic that deserves to be highlighted.

Improved home based management of diarrhoeal disease would result in dramatic impact of decreasing diarrhoeal related under five mortality. Your study stresses the fact that we are still not making adequate gains with this.

However, this manuscript has some major errors and omission of broad discussion topics and robust analysis. It will need major revision to bring up to standard for publication.

I have edited / made comments on your copy edit version of the manuscript and attached to this review for your consideration.

Further general comments:

- Please read through abstract and edit thoroughly to ensure no typos or grammatical errors. This is often the only section read by readers so needs to deliver information succinctly.

- it seems that there may have been some "cutting and pasting" from other studies that didn't belong in this manuscript which was rather concerning.

Eg. Paragraph 4 of introduction talks about combined zinc and ORS into a plastic pouch to enhance adherence as being the purpose of this study, but not mentioned again.

Table 1 is about cerebral palsy with completely different references to subsequent tables

- The discussion does not do the results justice. First 2 paragraphs are more suited in the background/introduction section. There is no robust dissection of the findings and how to interpret them. There is much heterogeneity in the results with very high I2 results, yet no discussion of this and what may be contributing to it. There is no discussion of the included study types that may have led to the heterogeneity. There was a subgroup analysis of the countries but no discussion of this. How are we meant to interpret these results - are they still clinically applicable or should we not have pooled them together?

- references: need to be reviewed carefully. The papers on cerebral palsy are all in here.

6. PLOS authors have the option to publish the peer review history of their article (what does this mean?). If published, this will include your full peer review and any attached files.

Reviewer #1: No

Reviewer #2: No

Reviewer #3: No

---

## [Author Response · Author response to Decision Letter 0]

6 Dec 2023

Point by point response

Dear Editor, 

Many thanks for letting us know such invaluable comments on our submission-"Knowledge, attitude and practice of home management of diarrhea among children in East Africa". Each points were very helpful for making our paper better and suitable for publication in your prestigious journal. As such, we have tried to amend the manuscript in light of all your and reviewers’ comments and the journal guideline. We included the point-to-point response to each comment as below. We have highlighted the track changes in the manuscript and attached as “Manuscript with track changes”. We have also attached a clean version manuscript. 

Dear Reviewers, 

Thank you so much for taking time to take a through look at our work and giving us such in-depth and invaluable feedback. All authors appreciated the way you commented the manuscript and we agreed with almost all of your comments and believe that we have amended the manuscript in light of all your comments. Many thanks again for your insightful feedback. We have indicated the point-by-point response for each of your comments below. 

Journal Requirements:

Editor comment 1. Please ensure that your manuscript meets PLOS ONE's style requirements, including those for file naming. The PLOS ONE style templates can be found at

Authors’ response: Thank you so much for providing us these important link; we have arranged the manuscript considering PLOS ONE's style requirements stated in the guideline

Editor comment 2. We noticed you have some minor occurrence of overlapping text with the following previous publication(s), which needs to be addressed:

-DOI: 10.1177/20503121221095727

-https://doi.org/10.1016/j.vhri.2021.03.005

In your revision ensure you cite all your sources (including your own works), and quote or rephrase any duplicated text outside the methods section. Further consideration is dependent on these concerns being addressed.

Authors’ response: Many thanks for raising such an important point. We have now rewrite the sections which have similarities with previous works, and we also cited all sources (including our own works).

 Editor comment 3. We note that you have indicated that data from this study are available upon request. PLOS only allows data to be available upon request if there are legal or ethical restrictions on sharing data publicly. 

For more information on unacceptable data access restrictions, please see http://journals.plos.org/plosone/s/data-availability#locunacceptable-data-access-restrictions.

b) If there are no restrictions, please upload the minimal anonymized data set necessary to replicate your study findings as either Supporting Information files or to a stable, public repository and provide us with the relevant URLs, DOIs, or accession numbers. For a list of acceptable repositories, please see http://journals.plos.org/plosone/s/dataavailability#loc-recommended-repositories.

Authors’ response: Thank you very much for letting us know these. We now make all data used in the analysis of this manuscript available. We included a statement indicating this. 

Editor comment PLOS requires an ORCID iD for the corresponding author in Editorial Manager on papers submitted after December 6th, 2016. Please ensure that you have an ORCID iD and that it is validated in Editorial Manager. To do this, go to ‘Update my Information’ (in the upper left-hand corner of the main menu), and click on the Fetch/Validate link next to the ORCID field. This will take you to the ORCID site and allow you to create a new iD or authenticate a pre-existing iD in Editorial Manager. Please see the following video for instructions on linking an ORCID iD to your Editorial Manager account: https://www.youtube.com/watch?v=_xcclfuvtxQ

Authors’ response: Many thanks for reminding this important issue. The corresponding author have now added his ORCID Id in both the system and the manuscript. 

Editor comment 5. Please amend the manuscript submission data (via Edit Submission) to include authors Gebremeskel Kibret Abebe and Freweyni Gebreegziabher Araya.

Authors’ response: We have amended authors name accordingly. 

Editor comment 6. Please amend your authorship list in your manuscript file to include authors Biruk Kibret Beletew and Biruk Gebreegziabher Beletew.

Authors’ response: We have amended authors name accordingly. 

Editor comment 7. Please include captions for your Supporting Information files at the end of your manuscript, and update any in-text citations to match accordingly. Please see our Supporting Information guidelines for more information: http://journals.plos.org/plosone/s/supporting-information.

Authors’ response: many thanks for the link on supporting Information guidelines. we have included captions on our supporting information files and updated the in-text citations accordingly. 

Additional Editor Comments:

Editor comment: The manuscript requires revision, since it has many errors and should be review-edited in proficient English for easy readability. 

Authors’ response: All authors re-edited the manuscript to remove the grammatical errors and improve the manuscript. Besides, we have send the manuscript to English language experts in our university and they have proof edited it.

Editor comment: The title of the manuscript should include the study type: Knowledge, attitude and practice of home management of diarrhea among children in East Africa; a Systematic Review and Meta-analysis and register your study protocol at PROSPERO. 

Authors’ response: thanks so much for raising such invaluable comment; we have amended the title adding the study type: a Systematic Review and Meta-analysis and we have submitted the protocol to PROSPERO for registration and it is under consideration. 

Editor comment: Additionally, please refer to the submission and publication criteria in the journal's guidelines for Systematic Reviews and Meta-Analysis.

Authors’ response: many thanks for directing us. We have checked PLOSE ONE journal's guidelines in general and for Systematic Reviews and Meta-Analysis in particular and amended the manuscript accordingly. 

Review Comments to the Author

Reviewer #1: 

Reviewer comment 1. The study characteristics table is on a different topic, and seemed to be inserted by mistake.

Authors’ response: Thank you so much for reminding us and please accept our heartfelt apology for such mistake. We have another manuscript about cerebral palsy. We have now amended the title of the table 1 and its content: Result: Page 7 Line 144-147

Reviewer comment 2. Subgroup analysis missing for Uganda and South Sudan, funnel plot missing from manuscript and supplementary materials.

Authors’ response: Many thanks for the comment. The articles by Nalubwama S et al(2021) from Uganda, Stephen B et al.(2016) from Uganda, and Aneeqa M etal(2022) from South Sudan did not report the knowledge level, but they did the practice level(Table 1). That is why we did not include these articles in the forest plot of subgroup analysis by country for knowledge level. Thus, not all included studies reported all the three outcomes (Knowledge, Attitude and Practice). 

Reviewer comment 3. All data included in the manuscript is fully available.

Authors’ response: yes, we have included all data we have used in the analysis of the manuscript and stated in the manuscript on data availability section. 

Reviewer comment 4. Not at all, authors need to revise their grammatical errors tremendously.

Authors’ response: All authors re-edited the manuscript to remove the grammatical errors and improve the manuscript. Besides, we have send the manuscript to English language experts in our university and they have proof edited it.

Reviewer #2: 

The authors conducted a systematic review of publications on the pooled knowledge, attitude and practices of caregivers of children on home-based diarrhea management in East African countries. The study determined that the values were 52.62%, 60% and 37.4%, respectively. Thus, the knowledge, attitude and practices of home-based management of diarrhea in East Africa were found to be low in contrast to some previous reports and despite WHO recommendation. 

Reviewer comment: Although, this is a somewhat important summary, it does not seem that the analysis contributes much (if any) novel knowledge or resolves any existing controversies/ inconsistencies in the field. 

Authors’ response: 

• The aim of this systematic review and meta-analysis was to assess the pooled estimate of knowledge, attitude and practice of home based management of diarrhea in East Africa.

• Inconsistent results: In East Africa different studies had been conducted regarding knowledge, attitude, and practice of home based management of diarrhea and results are variable (knowledge level ranges from 36.6% to 67%, attitude level 45.1% to 94.4%, and practice 12% to 58%).

• As per the investigators' knowledge, systematic reviews and meta-analyses to address these inconsistent results from East Africa are scarce. Introduction: Page 4 Line 69-74

Reviewer comment: Also, a thorough revision throughout the manuscript is needed to ensure more concise, proper and informative language/style. Currently, it is a struggle to read.

Authors’ response: All authors re-edited the manuscript to remove the grammatical errors and improve the manuscript. Besides, we have send the manuscript to English language experts in our university and they have proof edited it.

Specific comment

Reviewer comment: Please clarify why “Finally, 206 studies were screened for full-text review, and 15 articles with (n = 498406 patients) were included for the final analysis (Fig. 1).” (P8). What was the basis for non-selecting other 191 articles? Why were they excluded and only 15 were selected? 

Authors’ response: Thanks for raising such an important issue. The authors have now revised to make it easy to follow. “A total of 4394 studies were identified; 4380 from different databases and 14 from other sources. After duplication removed, a total of 2,431 articles remained (1963 removed by duplication). Finally, 206 studies were screened for full-text review, and 19 articles with (n = 498406 patients) were included for the final analysis (Fig. 1)”. Result: Page 7 Line 124-142

Reviewer comment: What is the meaning of ‘quality reasons’ and ‘Didn’t report outcome of interest’ shown in the diagram.

Authors’ response: the authors stated ‘quality reasons’ to mean Articles with poor quality (JBI score <4) Method: Page 6 Line 105-109, and ‘Didn’t report outcome of interest- to mean those articles which did not operationalize and report the level of knowledge, attitude and/or practice explicitly. Result: Figure 1 Page 7 

Reviewer comment: Some examples of errors, poor style, typos. Keep in mind that almost every sentence needs some level of revision.

Authors’ response: All authors re-edited the manuscript to remove the grammatical errors and improve the manuscript. Besides, we have send the manuscript to English language experts in our university and they have proof edited it.

Reviewer comment: P1: “Diarrheal disproportionately affects locations with poor access..” → “Diarrheal disease disproportionately affects locations with limited access..” Can’t say poor access. Same sentence needs to be corrected on P2.

Authors’ response: Many thanks for such through and in-depth comments. We have amended those comments as per your comments. Abstract: Page 1 Line 13

Reviewer comment: P1: “Factors of particular importance include care givers knowledge…”

Please revise throughout to avoid saying: “estimate the pooled estimates of knowledge”. Poor style. Please replace “care givers knowledge” with ‘caregiver knowledge’ throughout.

Authors’ response: Thanks for your comment. we have amended the manuscript throughout considering your comment. 

Reviewer comment: P2: “From the random-effects model analysis The pooled prevalence of good practice.” Do not randomly capitalize words inside a sentence.

Authors’ response: We thank you for raising such issue. We have now amended such typo issues. 

Reviewer comment: P2: “The level of good knowledge, attitude and practice of home based management of diarrhea in East Africa is found to low” → “The level of good knowledge, attitude and practice of home based management of diarrhea in East Africa is found to be low”

Authors’ response: Thanks so much. We have amended it. 

Reviewer comment: P3: “Sub-Saharan Africa countries take more than half of the global burden of under-five mortality.” → “Sub-Saharan African countries account more than half of the global burden of under-five mortality.”

Authors’ response: Amended. Introduction: Page 3 Line 45

Reviewer comment: P3: “Diarrhea in children can be managed at home before it becomes sever and problematic.” → “Diarrhea in children can be managed at home before it becomes severe and problematic.” I would also replace ‘problematic’ with ‘life threatening’

Authors’ response: Thanks. Rephrased. Introduction: Page 3 line 3

Reviewer comment: P3: “…often do not have access to formal healthcare”… What is ‘formal healthcare’? ‘Professional healthcare’

Authors’ response: Thanks. Rephrased. Introduction: Page 3 line 57

Reviewer comment: P3: “The level of home management practice of diarrhea is poor.” Please provide a reference to support this statement.

Authors’ response: this section has been revised and reference is included as per your comment Introduction: Page 3 line 56-57

Reviewer comment: P3: “. Similarly, their practice to use universal popular ORS in preventing dehydration due to diarrhea is also very low [3]” Whose is ‘their’? Who are you referring to with this pronoun?

Authors’ response: “Their” in this sentence is to refer caregivers. It is amended to make it easy to understand and follow. 

Reviewer comment: P4: “Family plays the major role in the treatment and surviving chance of a child with diarrhea.” → “Family plays the major role in the treatment and survival of children with diarrhea.”

Authors’ response: Thanks. amended. Introduction: Page 4 line 66

Reviewer comment: P4: “In Africa different studies have conducted regarding Knowledge, attitude, and practice on home based management of diarrhea and they lack consistency (knowledge ranges from 36.6% to 67%, attitude 45.1% to 94.4%, and practice 12% to 58%).” → “In Africa different studies have been conducted regarding the knowledge, attitude, and

practice of home-based management of diarrhea but they lack consistency (knowledge ranges from 36.6% to 67%, attitude 45.1% to 94.4%, and practice 12% to 58%).”

Unnecessary/imp[roper capitalization, improper use of conjunctions, missing articles and verbs are a big issues throughout the manuscript.

Authors’ response: We thank you for your interesting comments. We have amended it. Introduction: Page 4 line 70-72

Reviewer comment: P4: “As per the investigators knowledge there are no systematic review and meta-analysis done to address the inconsistence reports from Africa.” → “As per the investigators knowledge, no systematic review or meta-analysis was done to address this inconsistency (or these inconsistent data) from Africa.”

Authors’ response: Many thanks. Revised as per your comment. Introduction: Page 4 line 72-73

Reviewer comment: P5: “We searched these articles from the following databases: Cochrane library, Ovid platform (Medline, Embase, and Emcare), Google Scholar, CINAHL, PubMed, and institutional repositories in East Africa countries on 01/06/ 2023 G.

C” → “We searched these articles from the following databases: Cochrane library, Ovid platform (Medline, Embase, and Emcare), Google Scholar, CINAHL, PubMed, and institutional repositories in East African countries on 01/06/ 2023 G. C”

Authors’ response: Thanks. Amended Methods: Page 4 line 78-80

Reviewer comment: P8: Either omit ‘After duplication removed’ or “(1963 removed by duplication)”, having both in the sam sentence is redundant.

Authors’ response: Thank you so much for the comment. we have revised it. Results 7: Page 4 line 126-128

Reviewer comment: P17: “A concerted effort is needed to put an end to all avoidable infant and child deaths under the age of five in orderto achieve the sustainable development goals (SDG) aim of decreasing under-five mortality to 25 per 1000 live births[50].” → “A concerted effort is needed to put an end to all avoidable deaths of infants and children under the age of five in order to achieve the sustainable development goals (SDG) aim of decreasing under-five mortality to 25 per 1000 live births[50].” I would omit ‘under the age of five’ The aim should be to prevent all avoidable death for children of all ages.

Authors’ response: Thank you so much for the comment. we have revised it. Discussion: Page 17, 194-195

Reviewer #3: 

Reviewer comment: Well done on an important topic that deserves to be highlighted.

Improved home based management of diarrhoeal disease would result in dramatic impact of decreasing diarrhoeal related under five mortality. Your study stresses the fact that we are still not making adequate gains with this. However, this manuscript has some major errors and omission of broad discussion topics and robust analysis. It will need major revision to bring up to standard for publication. I have edited / made comments on your copy edit version of the manuscript and attached to this review for your consideration.

Further general comments:

Reviewer comment: - Please read through abstract and edit thoroughly to ensure no typos or grammatical errors. This is often the only section read by readers so needs to deliver information succinctly.

Authors’ response: All authors re-edited the manuscript to remove the grammatical errors and improve the manuscript. Besides, we have send the manuscript to English language experts in our university and they have proof edited it.

Reviewer comment: - it seems that there may have been some "cutting and pasting" from other studies that didn't belong in this manuscript which was rather concerning.

Eg. Paragraph 4 of introduction talks about combined zinc and ORS into a plastic pouch to enhance adherence as being the purpose of this study, but not mentioned again.

Authors’ response: Right you are! Please accept our big apology for including some unrelated sentences in this manuscript. We have other manuscript on related topic. Thanks, now we have removed all un necessary sentences and replaced them with more relevant one. We have now fixed it. 

Reviewer comment: Table 1 is about cerebral palsy with completely different references to subsequent tables

Authors’ response: Thank you so much for reminding us and please accept our heartfelt apology for such mistake. As stated, we have another manuscript about cerebral palsy. We have now amended the title of the table 1: Result: Page 7 Line 144-147

Reviewer comment: - The discussion does not do the results justice. First 2 paragraphs are more suited in the background/introduction section. There is no robust dissection of the findings and how to interpret them. There is much heterogeneity in the results with very high I2 results, yet no discussion of this and what may be contributing to it. There is no discussion of the included study types that may have led to the heterogeneity

Authors’ response: Thank you so much for raising such important issues; we have now revised the discussion section as per your comments. We have removed the first two paragraphs. We have also added possible explanations for the findings and the high heterogenicity. 

Reviewer comment: There was a subgroup analysis of the countries but no discussion of this. Authors’ response: Thanks for this important comment. We have now included discussion of the subgroup analysis result. 

Reviewer comment: How are we meant to interpret these results - are they still clinically applicable or should we not have pooled them together?

Authors’ response: Yes, the result is clinically important because the paper is on home-based management of diarrhea addressing the knowledge, attitude, and practice of care givers. Knowing the KAP of caregivers on home-based management of diarrhea have a direct link with reduction of under five mortality; as diarrhea is the most common cause of death in this age group specially in developing countries. 

Reviewer comment: - references: need to be reviewed carefully. The papers on cerebral palsy are all in here.

Authors’ response: please accept our apology for including unrelated references. As we have explained earlier it was by mistake that we included information about cerebral palsy from another work. We removed all unrelated information and updated the reference section. 

Major changes:

Reviewers comment 1: The manuscript mentions East Africa, however throughout the manuscript, authors have used the word Africa as a whole. This fails to clarify their intent behind writing. In the final analysis, authors include countries which are outside the boundaries of ‘East’ Africa, 

Authors’ Response: Thank you so much for such interesting comment. Sorry for such the mistake we did. The whole analysis was done based on included studies in East Africa. However, in the write up we have mixed with our study in Africa on cerebral pals; we have used that paper as a skeleton for the current study. Now everything is revised, and both the analysis and the write-up is based on the included studies from East Africa. Your comment is greatly appreciated. 

Reviewers comment 2: Authors have made a blinding mistake of inserting a study characteristics table which is on ‘Cerebral Palsy’ instead of the relevant topic at hand. 

Authors’ Response: Thank you so much for reminding us and please accept our heartfelt apology for such mistake. As stated, we have another manuscript about cerebral palsy. We have now amended the title of the table 1: Result: Page 7 Line 144-147

Reviewers comment 3: Authors have been reckless with their grammar, 80% of the time was spent correcting the grammar. There was no proof reading done before submission, and it shows. 

Authors’ Response: All authors re-edited the manuscript to remove the grammatical errors and improve the manuscript. Besides, we have sent the manuscript to English language experts in our university and they have proof edited it.

Reviewers comment 4: Figure 1 is also missing; the figures start from figure number 2.

Authors’ Response: Thanks. We have now included figure 1 in the text and figure. 

Reviewers comment 5: It was painful to see the amount of repetition this manuscript had. Authors were going in circles trying to mention just the mortality caused by the diarrhea. They failed to mention anything else. This manuscript does not talk about anything else. There was an interesting link between co-packaging zinc with ORS, however authors forgot about that and abandoned that just as abruptly as they introduced it in the article.

Authors’ Response: Please accept our big mistake, including some unrelated sentences in this manuscript; they are actually from our own related work. Thanks, now we have removed all un necessary sentences and replaced them with more relevant one. Right you are! now we have removed all un necessary sentences and replaced them with more relevant one. Thanks for your attention. 

Reviewers comment 6: Lack of funnel plot and subgroup analysis on South Sudan and Uganda.

Authors’ Response: Many thanks for the comment. The articles by Nalubwama S et al(2021) from Uganda, Stephen B et al.(2016) from Uganda, and Aneeqa M etal(2022) from South Sudan did not report the knowledge level, but they did the practice level(Table 1). That is why we did not include these articles in the forest plot of subgroup analysis by country for knowledge level. Thus, not all included studies reported all the three outcomes (Knowledge, Attitude and Practice). 

Reviewers comment 7: Discussion lacked the knowledge gap. Failed to discuss the significance of results, and did not link all the manuscript together.

Authors’ Response: Thanks for raising such an important issue. Now we have included information: Discussion: Page 17 line 191-195

Reviewers comment 8: No mention of limitations of study

Authors’ Response: Thank you so much we have now included the limitation of the study at the end of the discussion: Discussion: Page 18 line 228

Reviewers comment 9: Conclusion needs to be more descriptive as to what should be done.

Authors’ Response: Thanks, we have revised the conclusion section in light of your comments. Conclusion: Page 19 line 232-235

Reviewers comment 9: Minor changes: I have mentioned the rest of the comments within the manuscript, kindly look at them. The changes that were made by the reviewer were highlighted in yellow. Changes that need to be made are within notes marked as comments. In my opinion, the authors can improve and work on this interesting topic, however, they will need to work on it heavily, especially the purpose of what they write. They need to go over redundancy and stop quoting the same bit of information a thousand times.

Authors’ Response: Thanks, now we have removed all un necessary sentences and replaced them with more relevant one. Right you are! now we have removed all un necessary sentences and replaced them with more relevant one. Thanks for your attention.

---

## [Editor Report · Decision Letter 1]

18 Dec 2023

PONE-D-23-27980R1Knowledge, attitude and practice of home management of diarrhea among under five children in East Africa: A systematic Review and Meta-analysisPLOS ONE

Dear Dr. Abate,

Thank you for submitting your manuscript to PLOS ONE. After careful consideration, we feel that it has merit but does not fully meet PLOS ONE’s publication criteria as it currently stands. Therefore, we invite you to submit a revised version of the manuscript that addresses the points raised during the review process.

Please thoroughly revise the languages as numerous issues remain unresolved. The authors are encouraged to seek assistance from language experts for necessary amendments.

We look forward to receiving your revised manuscript.

Kind regards,

Wudneh Simegn, MSc

Academic Editor

PLOS ONE
---

## [Author Response · Author response to Decision Letter 1]

4 Jan 2024

Point by point response

Dear Editor, 

Many thanks for letting us know such invaluable comments on our submission-"Knowledge, attitude and practice of home management of diarrhea among children in East Africa". Each points were very helpful for making our paper better and suitable for publication in your prestigious journal. As such, we have tried to amend the manuscript in light of all your and reviewers’ comments and the journal guideline. We included the point-to-point response to each comment as below. We have highlighted the track changes in the manuscript and attached as “Manuscript with track changes”. We have also attached a clean version manuscript. 

Editor comment: Please thoroughly revise the languages as numerous issues remain unresolved. The authors are encouraged to seek assistance from language experts for necessary amendments.

Authors’ response: All authors re-edited the manuscript to remove the grammatical errors and improve the manuscript. Besides, we have send the manuscript to English language experts in our university and they have proof edited it. Thus, we believe now the manuscript has been greatly improved considering your and reviewers comments. 

Dear Reviewers, 

Thank you so much for taking time to take a through look at our work and giving us such in-depth and invaluable feedback. All authors appreciated the way you commented the manuscript and we agreed with almost all of your comments and believe that we have amended the manuscript in light of all your comments. Many thanks again for your insightful feedback. We have indicated the point-by-point response for each of your comments below. 

Journal Requirements:

Editor comment 1. Please ensure that your manuscript meets PLOS ONE's style requirements, including those for file naming. The PLOS ONE style templates can be found at

Authors’ response: Thank you so much for providing us these important link; we have arranged the manuscript considering PLOS ONE's style requirements stated in the guideline

Editor comment 2. We noticed you have some minor occurrence of overlapping text with the following previous publication(s), which needs to be addressed:

-DOI: 10.1177/20503121221095727

-https://doi.org/10.1016/j.vhri.2021.03.005

In your revision ensure you cite all your sources (including your own works), and quote or rephrase any duplicated text outside the methods section. Further consideration is dependent on these concerns being addressed.

Authors’ response: Many thanks for raising such an important point. We have now rewrite the sections which have similarities with previous works, and we also cited all sources (including our own works).

 Editor comment 3. We note that you have indicated that data from this study are available upon request. PLOS only allows data to be available upon request if there are legal or ethical restrictions on sharing data publicly. 

For more information on unacceptable data access restrictions, please see http://journals.plos.org/plosone/s/data-availability#locunacceptable-data-access-restrictions.

b) If there are no restrictions, please upload the minimal anonymized data set necessary to replicate your study findings as either Supporting Information files or to a stable, public repository and provide us with the relevant URLs, DOIs, or accession numbers. For a list of acceptable repositories, please see http://journals.plos.org/plosone/s/dataavailability#loc-recommended-repositories.

Authors’ response: Thank you very much for letting us know these. We now make all data used in the analysis of this manuscript available. We included a statement indicating this. 

Editor comment PLOS requires an ORCID iD for the corresponding author in Editorial Manager on papers submitted after December 6th, 2016. Please ensure that you have an ORCID iD and that it is validated in Editorial Manager. To do this, go to ‘Update my Information’ (in the upper left-hand corner of the main menu), and click on the Fetch/Validate link next to the ORCID field. This will take you to the ORCID site and allow you to create a new iD or authenticate a pre-existing iD in Editorial Manager. Please see the following video for instructions on linking an ORCID iD to your Editorial Manager account: https://www.youtube.com/watch?v=_xcclfuvtxQ

Authors’ response: Many thanks for reminding this important issue. The corresponding author have now added his ORCID Id in both the system and the manuscript. 

Editor comment 5. Please amend the manuscript submission data (via Edit Submission) to include authors Gebremeskel Kibret Abebe and Freweyni Gebreegziabher Araya.

Authors’ response: We have amended authors name accordingly. 

Editor comment 6. Please amend your authorship list in your manuscript file to include authors Biruk Kibret Beletew and Biruk Gebreegziabher Beletew.

Authors’ response: We have amended authors name accordingly. 

Editor comment 7. Please include captions for your Supporting Information files at the end of your manuscript, and update any in-text citations to match accordingly. Please see our Supporting Information guidelines for more information: http://journals.plos.org/plosone/s/supporting-information.

Authors’ response: many thanks for the link on supporting Information guidelines. we have included captions on our supporting information files and updated the in-text citations accordingly. 

Additional Editor Comments:

Editor comment: The manuscript requires revision, since it has many errors and should be review-edited in proficient English for easy readability. 

Authors’ response: All authors re-edited the manuscript to remove the grammatical errors and improve the manuscript. Besides, we have send the manuscript to English language experts in our university and they have proof edited it.

Editor comment: The title of the manuscript should include the study type: Knowledge, attitude and practice of home management of diarrhea among children in East Africa; a Systematic Review and Meta-analysis and register your study protocol at PROSPERO. 

Authors’ response: thanks so much for raising such invaluable comment; we have amended the title adding the study type: a Systematic Review and Meta-analysis and we have submitted the protocol to PROSPERO for registration and it is under consideration. 

Editor comment: Additionally, please refer to the submission and publication criteria in the journal's guidelines for Systematic Reviews and Meta-Analysis.

Authors’ response: many thanks for directing us. We have checked PLOSE ONE journal's guidelines in general and for Systematic Reviews and Meta-Analysis in particular and amended the manuscript accordingly. 

Review Comments to the Author

Reviewer #1: 

Reviewer comment 1. The study characteristics table is on a different topic, and seemed to be inserted by mistake.

Authors’ response: Thank you so much for reminding us and please accept our heartfelt apology for such mistake. We have another manuscript about cerebral palsy. We have now amended the title of the table 1 and its content: Result: Page 7 Line 144-147

Reviewer comment 2. Subgroup analysis missing for Uganda and South Sudan, funnel plot missing from manuscript and supplementary materials.

Authors’ response: Many thanks for the comment. The articles by Nalubwama S et al(2021) from Uganda, Stephen B et al.(2016) from Uganda, and Aneeqa M etal(2022) from South Sudan did not report the knowledge level, but they did the practice level(Table 1). That is why we did not include these articles in the forest plot of subgroup analysis by country for knowledge level. Thus, not all included studies reported all the three outcomes (Knowledge, Attitude and Practice). 

Reviewer comment 3. All data included in the manuscript is fully available.

Authors’ response: yes, we have included all data we have used in the analysis of the manuscript and stated in the manuscript on data availability section. 

Reviewer comment 4. Not at all, authors need to revise their grammatical errors tremendously.

Authors’ response: All authors re-edited the manuscript to remove the grammatical errors and improve the manuscript. Besides, we have send the manuscript to English language experts in our university and they have proof edited it.

Reviewer #2: 

The authors conducted a systematic review of publications on the pooled knowledge, attitude and practices of caregivers of children on home-based diarrhea management in East African countries. The study determined that the values were 52.62%, 60% and 37.4%, respectively. Thus, the knowledge, attitude and practices of home-based management of diarrhea in East Africa were found to be low in contrast to some previous reports and despite WHO recommendation. 

Reviewer comment: Although, this is a somewhat important summary, it does not seem that the analysis contributes much (if any) novel knowledge or resolves any existing controversies/ inconsistencies in the field. 

Authors’ response: 

• The aim of this systematic review and meta-analysis was to assess the pooled estimate of knowledge, attitude and practice of home based management of diarrhea in East Africa.

• Inconsistent results: In East Africa different studies had been conducted regarding knowledge, attitude, and practice of home based management of diarrhea and results are variable (knowledge level ranges from 36.6% to 67%, attitude level 45.1% to 94.4%, and practice 12% to 58%).

• As per the investigators' knowledge, systematic reviews and meta-analyses to address these inconsistent results from East Africa are scarce. Introduction: Page 4 Line 69-74

Reviewer comment: Also, a thorough revision throughout the manuscript is needed to ensure more concise, proper and informative language/style. Currently, it is a struggle to read.

Authors’ response: All authors re-edited the manuscript to remove the grammatical errors and improve the manuscript. Besides, we have send the manuscript to English language experts in our university and they have proof edited it.

Specific comment

Reviewer comment: Please clarify why “Finally, 206 studies were screened for full-text review, and 15 articles with (n = 498406 patients) were included for the final analysis (Fig. 1).” (P8). What was the basis for non-selecting other 191 articles? Why were they excluded and only 15 were selected? 

Authors’ response: Thanks for raising such an important issue. The authors have now revised to make it easy to follow. “A total of 4394 studies were identified; 4380 from different databases and 14 from other sources. After duplication removed, a total of 2,431 articles remained (1963 removed by duplication). Finally, 206 studies were screened for full-text review, and 19 articles with (n = 498406 patients) were included for the final analysis (Fig. 1)”. Result: Page 7 Line 124-142

Reviewer comment: What is the meaning of ‘quality reasons’ and ‘Didn’t report outcome of interest’ shown in the diagram.

Authors’ response: the authors stated ‘quality reasons’ to mean Articles with poor quality (JBI score <4) Method: Page 6 Line 105-109, and ‘Didn’t report outcome of interest- to mean those articles which did not operationalize and report the level of knowledge, attitude and/or practice explicitly. Result: Figure 1 Page 7 

Reviewer comment: Some examples of errors, poor style, typos. Keep in mind that almost every sentence needs some level of revision.

Authors’ response: All authors re-edited the manuscript to remove the grammatical errors and improve the manuscript. Besides, we have send the manuscript to English language experts in our university and they have proof edited it.

Reviewer comment: P1: “Diarrheal disproportionately affects locations with poor access..” → “Diarrheal disease disproportionately affects locations with limited access..” Can’t say poor access. Same sentence needs to be corrected on P2.

Authors’ response: Many thanks for such through and in-depth comments. We have amended those comments as per your comments. Abstract: Page 1 Line 13

Reviewer comment: P1: “Factors of particular importance include care givers knowledge…”

Please revise throughout to avoid saying: “estimate the pooled estimates of knowledge”. Poor style. Please replace “care givers knowledge” with ‘caregiver knowledge’ throughout.

Authors’ response: Thanks for your comment. we have amended the manuscript throughout considering your comment. 

Reviewer comment: P2: “From the random-effects model analysis The pooled prevalence of good practice.” Do not randomly capitalize words inside a sentence.

Authors’ response: We thank you for raising such issue. We have now amended such typo issues. 

Reviewer comment: P2: “The level of good knowledge, attitude and practice of home based management of diarrhea in East Africa is found to low” → “The level of good knowledge, attitude and practice of home based management of diarrhea in East Africa is found to be low”

Authors’ response: Thanks so much. We have amended it. 

Reviewer comment: P3: “Sub-Saharan Africa countries take more than half of the global burden of under-five mortality.” → “Sub-Saharan African countries account more than half of the global burden of under-five mortality.”

Authors’ response: Amended. Introduction: Page 3 Line 45

Reviewer comment: P3: “Diarrhea in children can be managed at home before it becomes sever and problematic.” → “Diarrhea in children can be managed at home before it becomes severe and problematic.” I would also replace ‘problematic’ with ‘life threatening’

Authors’ response: Thanks. Rephrased. Introduction: Page 3 line 3

Reviewer comment: P3: “…often do not have access to formal healthcare”… What is ‘formal healthcare’? ‘Professional healthcare’

Authors’ response: Thanks. Rephrased. Introduction: Page 3 line 57

Reviewer comment: P3: “The level of home management practice of diarrhea is poor.” Please provide a reference to support this statement.

Authors’ response: this section has been revised and reference is included as per your comment Introduction: Page 3 line 56-57

Reviewer comment: P3: “. Similarly, their practice to use universal popular ORS in preventing dehydration due to diarrhea is also very low [3]” Whose is ‘their’? Who are you referring to with this pronoun?

Authors’ response: “Their” in this sentence is to refer caregivers. It is amended to make it easy to understand and follow. 

Reviewer comment: P4: “Family plays the major role in the treatment and surviving chance of a child with diarrhea.” → “Family plays the major role in the treatment and survival of children with diarrhea.”

Authors’ response: Thanks. amended. Introduction: Page 4 line 66

Reviewer comment: P4: “In Africa different studies have conducted regarding Knowledge, attitude, and practice on home based management of diarrhea and they lack consistency (knowledge ranges from 36.6% to 67%, attitude 45.1% to 94.4%, and practice 12% to 58%).” → “In Africa different studies have been conducted regarding the knowledge, attitude, and

practice of home-based management of diarrhea but they lack consistency (knowledge ranges from 36.6% to 67%, attitude 45.1% to 94.4%, and practice 12% to 58%).”

Unnecessary/imp[roper capitalization, improper use of conjunctions, missing articles and verbs are a big issues throughout the manuscript.

Authors’ response: We thank you for your interesting comments. We have amended it. Introduction: Page 4 line 70-72

Reviewer comment: P4: “As per the investigators knowledge there are no systematic review and meta-analysis done to address the inconsistence reports from Africa.” → “As per the investigators knowledge, no systematic review or meta-analysis was done to address this inconsistency (or these inconsistent data) from Africa.”

Authors’ response: Many thanks. Revised as per your comment. Introduction: Page 4 line 72-73

Reviewer comment: P5: “We searched these articles from the following databases: Cochrane library, Ovid platform (Medline, Embase, and Emcare), Google Scholar, CINAHL, PubMed, and institutional repositories in East Africa countries on 01/06/ 2023 G.

C” → “We searched these articles from the following databases: Cochrane library, Ovid platform (Medline, Embase, and Emcare), Google Scholar, CINAHL, PubMed, and institutional repositories in East African countries on 01/06/ 2023 G. C”

Authors’ response: Thanks. Amended Methods: Page 4 line 78-80

Reviewer comment: P8: Either omit ‘After duplication removed’ or “(1963 removed by duplication)”, having both in the sam sentence is redundant.

Authors’ response: Thank you so much for the comment. we have revised it. Results 7: Page 4 line 126-128

Reviewer comment: P17: “A concerted effort is needed to put an end to all avoidable infant and child deaths under the age of five in orderto achieve the sustainable development goals (SDG) aim of decreasing under-five mortality to 25 per 1000 live births[50].” → “A concerted effort is needed to put an end to all avoidable deaths of infants and children under the age of five in order to achieve the sustainable development goals (SDG) aim of decreasing under-five mortality to 25 per 1000 live births[50].” I would omit ‘under the age of five’ The aim should be to prevent all avoidable death for children of all ages.

Authors’ response: Thank you so much for the comment. we have revised it. Discussion: Page 17, 194-195

Reviewer #3: 

Reviewer comment: Well done on an important topic that deserves to be highlighted.

Improved home based management of diarrhoeal disease would result in dramatic impact of decreasing diarrhoeal related under five mortality. Your study stresses the fact that we are still not making adequate gains with this. However, this manuscript has some major errors and omission of broad discussion topics and robust analysis. It will need major revision to bring up to standard for publication. I have edited / made comments on your copy edit version of the manuscript and attached to this review for your consideration.

Further general comments:

Reviewer comment: - Please read through abstract and edit thoroughly to ensure no typos or grammatical errors. This is often the only section read by readers so needs to deliver information succinctly.

Authors’ response: All authors re-edited the manuscript to remove the grammatical errors and improve the manuscript. Besides, we have send the manuscript to English language experts in our university and they have proof edited it.

Reviewer comment: - it seems that there may have been some "cutting and pasting" from other studies that didn't belong in this manuscript which was rather concerning.

Eg. Paragraph 4 of introduction talks about combined zinc and ORS into a plastic pouch to enhance adherence as being the purpose of this study, but not mentioned again.

Authors’ response: Right you are! Please accept our big apology for including some unrelated sentences in this manuscript. We have other manuscript on related topic. Thanks, now we have removed all un necessary sentences and replaced them with more relevant one. We have now fixed it. 

Reviewer comment: Table 1 is about cerebral palsy with completely different references to subsequent tables

Authors’ response: Thank you so much for reminding us and please accept our heartfelt apology for such mistake. As stated, we have another manuscript about cerebral palsy. We have now amended the title of the table 1: Result: Page 7 Line 144-147

Reviewer comment: - The discussion does not do the results justice. First 2 paragraphs are more suited in the background/introduction section. There is no robust dissection of the findings and how to interpret them. There is much heterogeneity in the results with very high I2 results, yet no discussion of this and what may be contributing to it. There is no discussion of the included study types that may have led to the heterogeneity

Authors’ response: Thank you so much for raising such important issues; we have now revised the discussion section as per your comments. We have removed the first two paragraphs. We have also added possible explanations for the findings and the high heterogenicity. 

Reviewer comment: There was a subgroup analysis of the countries but no discussion of this. Authors’ response: Thanks for this important comment. We have now included discussion of the subgroup analysis result. 

Reviewer comment: How are we meant to interpret these results - are they still clinically applicable or should we not have pooled them together?

Authors’ response: Yes, the result is clinically important because the paper is on home-based management of diarrhea addressing the knowledge, attitude, and practice of care givers. Knowing the KAP of caregivers on home-based management of diarrhea have a direct link with reduction of under five mortality; as diarrhea is the most common cause of death in this age group specially in developing countries. 

Reviewer comment: - references: need to be reviewed carefully. The papers on cerebral palsy are all in here.

Authors’ response: please accept our apology for including unrelated references. As we have explained earlier it was by mistake that we included information about cerebral palsy from another work. We removed all unrelated information and updated the reference section. 

Major changes:

Reviewers comment 1: The manuscript mentions East Africa, however throughout the manuscript, authors have used the word Africa as a whole. This fails to clarify their intent behind writing. In the final analysis, authors include countries which are outside the boundaries of ‘East’ Africa, 

Authors’ Response: Thank you so much for such interesting comment. Sorry for such the mistake we did. The whole analysis was done based on included studies in East Africa. However, in the write up we have mixed with our study in Africa on cerebral pals; we have used that paper as a skeleton for the current study. Now everything is revised, and both the analysis and the write-up is based on the included studies from East Africa. Your comment is greatly appreciated. 

Reviewers comment 2: Authors have made a blinding mistake of inserting a study characteristics table which is on ‘Cerebral Palsy’ instead of the relevant topic at hand. 

Authors’ Response: Thank you so much for reminding us and please accept our heartfelt apology for such mistake. As stated, we have another manuscript about cerebral palsy. We have now amended the title of the table 1: Result: Page 7 Line 144-147

Reviewers comment 3: Authors have been reckless with their grammar, 80% of the time was spent correcting the grammar. There was no proof reading done before submission, and it shows. 

Authors’ Response: All authors re-edited the manuscript to remove the grammatical errors and improve the manuscript. Besides, we have sent the manuscript to English language experts in our university and they have proof edited it.

Reviewers comment 4: Figure 1 is also missing; the figures start from figure number 2.

Authors’ Response: Thanks. We have now included figure 1 in the text and figure. 

Reviewers comment 5: It was painful to see the amount of repetition this manuscript had. Authors were going in circles trying to mention just the mortality caused by the diarrhea. They failed to mention anything else. This manuscript does not talk about anything else. There was an interesting link between co-packaging zinc with ORS, however authors forgot about that and abandoned that just as abruptly as they introduced it in the article.

Authors’ Response: Please accept our big mistake, including some unrelated sentences in this manuscript; they are actually from our own related work. Thanks, now we have removed all un necessary sentences and replaced them with more relevant one. Right you are! now we have removed all un necessary sentences and replaced them with more relevant one. Thanks for your attention. 

Reviewers comment 6: Lack of funnel plot and subgroup analysis on South Sudan and Uganda.

Authors’ Response: Many thanks for the comment. The articles by Nalubwama S et al(2021) from Uganda, Stephen B et al.(2016) from Uganda, and Aneeqa M etal(2022) from South Sudan did not report the knowledge level, but they did the practice level(Table 1). That is why we did not include these articles in the forest plot of subgroup analysis by country for knowledge level. Thus, not all included studies reported all the three outcomes (Knowledge, Attitude and Practice). 

Reviewers comment 7: Discussion lacked the knowledge gap. Failed to discuss the significance of results, and did not link all the manuscript together.

Authors’ Response: Thanks for raising such an important issue. Now we have included information: Discussion: Page 17 line 191-195

Reviewers comment 8: No mention of limitations of study

Authors’ Response: Thank you so much we have now included the limitation of the study at the end of the discussion: Discussion: Page 18 line 228

Reviewers comment 9: Conclusion needs to be more descriptive as to what should be done.

Authors’ Response: Thanks, we have revised the conclusion section in light of your comments. Conclusion: Page 19 line 232-235

Reviewers comment 9: Minor changes: I have mentioned the rest of the comments within the manuscript, kindly look at them. The changes that were made by the reviewer were highlighted in yellow. Changes that need to be made are within notes marked as comments. In my opinion, the authors can improve and work on this interesting topic, however, they will need to work on it heavily, especially the purpose of what they write. They need to go over redundancy and stop quoting the same bit of information a thousand times.

Authors’ Response: Thanks, now we have removed all un necessary sentences and replaced them with more relevant one. Right you are! now we have removed all un necessary sentences and replaced them with more relevant one. Thanks for your attention.

---

## [Editor Report · Decision Letter 2]

31 Jan 2024

Knowledge, attitude and practice of home management of diarrhea among under five children in East Africa: A systematic Review and Meta-analysis

PONE-D-23-27980R2

Dear Dr. Abate,

We’re pleased to inform you that your manuscript has been judged scientifically suitable for publication and will be formally accepted for publication once it meets all outstanding technical requirements.

Kind regards,

Wudneh Simegn, MSc

Academic Editor

PLOS ONE
---

## [Editor Report · Acceptance letter]

16 Feb 2024

PONE-D-23-27980R2 

PLOS ONE

Dear Dr. Abate, 

I'm pleased to inform you that your manuscript has been deemed suitable for publication in PLOS ONE. Congratulations! Your manuscript is now being handed over to our production team.

Kind regards, 

on behalf of

Dr. Wudneh Simegn 

Academic Editor

PLOS ONE